# The Response of Different-Levels Public Hospitals to Regional Global Budget with a Floating Payment System: Evidence from China

**DOI:** 10.3390/ijerph192315507

**Published:** 2022-11-23

**Authors:** Li Xiang, Zhengdong Zhong, Junnan Jiang

**Affiliations:** 1School of Medicine and Health Management, Huazhong University of Science and Technology, Wuhan 430030, China; 2School of Public Administration, Zhongnan University of Economics and Law, 182th South Lake Avenue, Wuhan 430073, China

**Keywords:** regional global budget, floating payment system, response of different-levels hospitals, transferring costs

## Abstract

Background: Regional Global Budget with a Floating Payment System (RGB-FPS) is a global budget widely used in medical insurance payments. However, existing studies on hospitals’ responses to RGB-FPS have limitations. First, existing studies have paid little attention to RGB-FPS’s macro effects. Theoretical studies did not analyze differences between different levels of hospitals. Secondly, studies did not reveal whether RGB-FPS has the same impact on the public-hospital-dominated market. Methods: First, we refine the research hypotheses through theoretical analysis. We then test the hypotheses empirically through interrupted time series analysis. Results: Theoretical analysis found that small hospitals were easier to transfer costs. The empirical analysis found that after RGB-FPS, the proportion of inpatients (PI)and the average times of inpatients in large hospitals increased (*p* < 0.001), and the proportion of non-reimbursable expenses (PNE) remained stable (*p* > 0.05). PI in secondary hospitals decreased (*p* < 0.01), and PNE increased (*p* < 0.01). PI in the primary hospital decreased (*p* < 0.05), and PNE increased (*p* < 0.001). Conclusion: This study verifies theoretically and empirically that large hospitals are easier to increase service volume and small hospitals are easier to transfer costs under the influence of RGB-FPS. Chinese public hospitals’ response to RGB-FPS is similar to that of private hospitals.

## 1. Introduction

Healthcare expenditures tend to increase worldwide. The increase in healthcare spending has alarmed governments [1,2]. To control healthcare costs, policymakers are gradually replacing post-payment systems with anticipatory payment systems [3]. The global budget (GB) is one of the successful anticipatory payment systems. However, despite the experience gained over the last three decades in developed countries, academic research on GB has not reached a consensus except for cost control and has limited recognition of the impact of GB on hospitals’ performance [4].

There are major differences in the implementation of GB, so Bradley defined three mechanisms of GB in their study [4]. The first mechanism is to determine expenditure by adjusting the price of services to reflect changes in the number of services (namely, the Regional Global Budget with a Floating Payment System, RGB-FPS). Under this mechanism, the GB is usually the total amount of all healthcare providers in a region and is not broken down into individual hospitals. At the same time, RGB-FPS can integrate other payment methods, such as Diagnosis Related Groups (DRG), Per diem, by disease, etc. The second and third mechanisms allocate fixed amounts of funds to health plans and providers, respectively, and the difference between the allocated budget and actual expenditures can be retained as profit or absorbed as a loss. The second and third mechanisms are widely used in the United States and many other developed countries [5,6,7,8]. Due to limited resources and weak regulatory capacity, RGB-FPS has been attempted to improve hospital behavior in recent years in low—and middle-income countries or areas, such as Thailand, Taiwan of China, and the mainland of China [4,9]. Compared to the other two GB mechanisms, there are few studies on RGB-FPS. Therefore, this study focuses on RGB-FPS.

The existing researches on RGB-FPS have some limitations. First, many studies focus on the micro performance of diseases but less on the macro performance of hospitals after the implementation of RGB-FPS [10,11,12,13,14,15,16,17,18]. Theoretical and empirical studies have revealed that RGB-FPS may lead hospitals to increase service volume by attracting patients from other hospitals and hospitalization decomposition [4,9]. However, empirical studies have found that large hospitals have a stronger ability to increase service volume, but theoretical studies do not discuss hospitals at different levels separately as empirical analyses [4,16,17]. So does RGB-FPS have different effects on hospitals at different levels? Do small hospitals take other actions when they are less able to increase service volume? Research is still lacking. Second, there is little research on the macro-impact of RGB-FPS on public hospitals. The study found that for-profit hospitals had a stronger response to RGB-FPS [19]. Although empirical studies have macroscopically analyzed the effect of RGB-FPS on hospitals, private for-profit hospitals account for more than 80% of the research [4]. China is vigorously promoting RGB-FPS. Will RGB-FPS have the same impact on a public-hospital-dominated market like China? There are few corresponding studies. 

Therefore, in this study, we introduced the RGB-FPS system by taking an example from City A. We conducted this research based on two hypotheses: To gain the maximum financial benefits: (1) large hospitals are more likely to increase service volume; (2) small hospitals are more likely to transfer costs. First, we theoretically investigated the changes and associated factors of service volume and transferring costs between large and small hospitals. We then evaluated the performance changes of the pilot hospitals in city A before and after the implementation of RGB-FPS in urban areas using interrupted time series analysis (ITSA) to test our hypotheses. The findings of this study might provide a basis for the implementation of RGB-FPS in China or other countries with large unmet responses to medical insurance payment methods.

## 2. Institutional Background—City A’s RGB-FPS

China has begun to promote the RGB-FPS nationwide [20,21]. As one of the demonstration cities, city A implemented the RGB-FPS earlier, and the effect of RGB-FPS in city A may be more obvious. Therefore, this article chooses city A as the research sample.

Inappropriate incentives for healthcare providers are a well-known problem in China [22]. Before 2017, the medical services in city A implemented a flat-rate system dominated by fee-for-service (FFS), which led to the overuse of drug prescriptions and diagnostic testing [23].

Although each hospital had an expenditure limit, the control effect was limited. The reason is that hospitals are not rewarded if they control their actual expenditure so that the expenditure is less than the limit. The best strategy for hospitals is to meet or exceed the expenditure limit, aiming for an increase in annual expenditure limits next year [9]. In 2014, the current income and expenditure of the medical insurance fund in city A showed a deficit. Since 2012, the average annual growth rate of medical insurance fund expenditure in city A has been about 14%.

To control the rising medical cost, city A carried out information system construction, supervision policy, and other supporting policies related to RGB-FPS in 7 pilot hospitals in July 2016. City A fully initiated implementation of RGB-FPS in all hospitals in urban areas in July 2017. (The cost and pricing system for RGB-FPS is shown in Figure 1).

City A’s RGB-FPS payment reform is purely supply-side reform without any reform on the demand side. So patients are still charged by FFS after the RGB-FPS reform. Since 2012, city A has merged the medical insurance for urban and rural residents, and RGB-FPS covers both urban and rural areas. Each RGB-FPS group will be allocated a certain number of points to reflect the group’s resource consumption. The RGB-FPS calculated the resource consumption of DRG, long-term chronic diseases, and special complex diseases according to the historical cost of DRG group cost, historical Per-diem cost, and service cost. The average cost of all diseases was used to set the number of points for each disease. Therefore, the RGB-FPS of city A covers almost all hospitalized cases in the urban area.

The point volume of each RGB-FPS group was calculated based on hospitalization cost data from January 2015 to June 2017 (The point volume is dynamically updated annually). The point volume for each RGB-FPS group is determined based on the RGB-FPS group’s average hospitalization cost and the urban areas’ average hospitalization cost.
$$\mathrm{The}\;\mathrm{point}\;\mathrm{volume}\;\mathrm{for}\;\mathrm{each}\;\mathrm{RGB}-\mathrm{FPS}=\frac{\mathrm{the}\;\mathrm{average}\;\mathrm{hospitalization}\;\cos t\;\mathrm{of}\;\mathrm{the}\;\mathrm{RGB}-\mathrm{FPS}\;\mathrm{group}}{\mathrm{the}\;\mathrm{average}\;\cos t\;\mathrm{of}\;\mathrm{hospitalization}\;\mathrm{in}\;\mathrm{urban}\;\mathrm{areas}}$$

The relative value (point volume) of each RGB-FPS group is fixed, but the actual monetary value is determined ex-post based on the point value (PV).
$$\mathrm{Point}\;\mathrm{Value}\;(\mathrm{PV})=\frac{\mathrm{the}\;\mathrm{total}\;\mathrm{amount}\;\mathrm{of}\;\mathrm{the}\;\mathrm{regional}\;\mathrm{global}\;\mathrm{budget}}{\mathrm{the}\;\mathrm{total}\;\mathrm{points}\;\mathrm{volume}\;\mathrm{of}\;\mathrm{all}\;\mathrm{urban}\;\mathrm{hospitals}}$$

The annual reimbursement amount of each hospital by the medical insurance management organization depends on its service volume. It also depends on the volume of services at other hospitals. The medical insurance income of the hospital is the sum of the medical insurance income for different diseases.
$$\mathrm{The}\;\mathrm{hospital}’s\;\mathrm{medical}\;\mathrm{insurance}\;\mathrm{income}=\sum_{\mathrm{group}\;1}^{\mathrm{group}\;n}(\mathrm{the}\;\mathrm{assigned}\;\mathrm{point}\;\mathrm{volume})\;\times \;\mathrm{PV}\;\times \;\mathrm{CC}$$

## 3. Theoretical Analysis

### 3.1. Existing Research on Hospital Behavior Analysis Theory under RGB-FPS Mechanism

The theory of common-property resources and game theory is often used in the effective analysis of RGB-FPS [24,25], The funds in RGB-FPS are essentially common-property resources. Common-property resources often suffer from overcrowding or overuse [26]. In this framework, healthcare providers share a limited RGB-FPS budget. However, hospitals often choose high-service volume strategies to ensure market share tends to be safer in the limited GB pool because they cannot predict what measures other hospitals are taking [27]. When multiple hospitals decide to increase their service volume, PV decreases, and the medical insurance funds obtained by different hospitals do not increase or even decrease. This situation is known as the prisoner’s dilemma in a non-cooperative game.

Many theoretical and empirical studies have verified the improvement in service volume. However, Bradley pointed out through game theory analysis that hospitals also have the behavior of shifting costs to high-profit services [4]. Shou-Hsia also proposed that hospitals may transfer costs to increase the out-of-pocket expenses of patients, but the study’s empirical findings were not significant [12]. Hsueh also points out that there may be transfer costs in dentistry that can lead to out-of-pocket expenses of total hospital costs up to 20% to 70% [10]. This suggests that RGB-FPS may lead hospitals to transfer costs.

In reality, large hospitals and small hospitals have different characteristics in personnel, equipment, technology, and other resources. According to the resource-based view [28,29], the profitability of large hospitals is stronger than that of small hospitals when large hospitals have resource advantages. Therefore, RGB-FPS may have different effects on different-level hospitals. Large hospitals are more likely to increase their market share by improving service volume, which has been confirmed by empirical studies [4,16,17]. However, in the study of transfer costs, the research does not divide the hospital into different levels to discuss in the theoretical analysis. Is there any difference between small hospitals and large hospitals in terms of cost transfer behavior after the implementation of RGB-FPS? Studies have not shed much light on this issue.

### 3.2. Theoretical Analysis of Different Types of Hospital Behaviors under RGB-FPS Mechanism

Based on the findings of previous studies, this study proposes two initial hypotheses about the impact of RGB-FPS:(1) In addition to increasing service volume, the hospital has the behavior of transferring costs. (2) Different-level hospitals may have different responses to the two behaviors. We first tested our hypotheses with a brief theoretical analysis. We compared the differences between different-levels hospitals in the theoretical analysis to theoretically verify that the behaviors (increasing service volume and transferring costs) of different-level hospitals may be different.

We put the differences between different-level hospitals into the theoretical analysis. We started with the hypothesis that there are two hospitals in the market, large hospitals (no.1) and small hospitals (no.2). Before the reform of RGB-FPS, the total service volume of the market is N ($N_{1}$ for large hospitals, $N_{2}$ for small hospitals, and ${N=N}_{1}{+N}_{2}$). After the reform of RGB-FPS, it is assumed that different hospitals may increase service volume or transfer costs. The service quantity of large hospitals is $n_{1}$, and the income of transferring costs is $T_{1}$. The service quantity of small hospitals is $n_{2}$, and the income of transferring costs is $T_{2}$. Then after the reform of RGB-FPS, the total service volume is $N_{1}{+n}_{1}{+N}_{2}{+n}_{2}$. The total medical insurance fund is S (large hospitals occupy $S_{1}$, small hospitals occupy $S_{2}$, ${S=S}_{1}{+S}_{2}$). The total benefits of large and small hospitals from increasing service volume or transferring expenses are $X_{1}$ and $X_{2}$, respectively. So:(1)$$S_{1}=\frac{N_{1}{+n}_{1}}{N_{1}{+n}_{1}{+N}_{2}{+n}_{2}}\times S$$
(2)$$S_{2}=\frac{N_{2}{+n}_{2}}{N_{1}{+n}_{1}{+N}_{2}{+n}_{2}}\times S$$
(3)$$X_{1}{=S}_{1}{+T}_{1}=\frac{1}{1+\frac{N_{2}{+n}_{2}}{N_{1}{+n}_{1}}}{\times S+T}_{1}$$
(4)$$X_{2}{=S}_{2}{+T}_{2}=\frac{1}{1+\frac{N_{1}{+n}_{1}}{N_{2}{+n}_{2}}}{\times S+T}_{2}$$

Transferring costs may increase patients’ out-of-pocket expenses and may lead to patients’ negative evaluation of the hospital, so large hospitals are less likely to choose to increase $T_{1}$. Based on previous research, the large hospital increases the service quantity more obviously, and the service quantity base is bigger because of the resource advantage. Therefore, we can calculate the following formula: $N_{1}{+n}_{1}$≫$N_{2}{+n}_{2}$, $0<\frac{N_{2}{+n}_{2}}{N_{1}{+n}_{1}}\ll 1$, $\frac{N_{1}{+n}_{1}}{N_{2}{+n}_{2}}\gg 1$. As $n_{1}$ becomes larger, $S_{1}\to S$ and $S_{2}\to 0$, so $X_{2}\to T_{2}$, which means that small hospitals have lower and lower income from the medical insurance fund. Since patients are still charged by FFS, small hospitals will increase $T_{2}$. by shifting costs to maintain revenue $X_{2}$.

Based on the above theoretical analysis, we improve two hypotheses: To gain the maximum financial benefits: (1) large hospitals are more likely to increase service volume; (2) small hospitals are more likely to transfer costs. Next, we use empirical analysis to verify our hypotheses.

## 4. Empirical Analysis

### 4.1. Materials

The districts of City A are over 20 streets and 18 towns. The total population of the urban area was 1,015,500 in 2018, and the per capita GDP was 76,044.2 yuan ($11,487).

We focused on the seven pilot hospitals (All public hospitals), which are more representative. As these seven pilot hospitals account for more than 80% of the total medical insurance fund in the urban area. Meanwhile, in July 2017, these seven pilot hospitals were not affected by other policies except RGB-FPS.

The data of this study come from the medical insurance management institution of city A. The monthly data are exported from the medical insurance management information system by city A medical insurance management agency.

### 4.2. Outcome Variables

The outcome variables were increasing service volume and transferring costs. The increasing service volume includes the proportion of inpatients (PI) and the average times of inpatients (ATI), and transferring costs include the proportion of non-reimbursable expenses (PNE).

The overprovision of service volume is the main impact of RGB-FPS, which has been confirmed by many studies in Taiwan and mainland China [4,9,12,30]. For overprovision of services, general studies use the absolute value of service volume to analyze whether hospitals are overserving [4,7,8]. Under the RGB-FPS mechanism, the overprovision of hospital services is usually done in two ways: siphoning patients of small hospitals and decompressing hospitalization [4,9]. It is difficult to use absolute values to analyze whether there are two cases. At the same time, the service volume of the hospital is not only affected by RGB-FPS but also by natural growth factors [4]. Therefore, this study chooses PI in different hospitals to analyze whether large hospitals attract patients from small hospitals, which can exclude the influence of resource advantages and patient preferences to a certain extent. ATI was used to analyze the presence of hospitalization decomposition. Based on the theoretical hypotheses, if large hospitals have more advantages in providing excessive medical services, the PI in large hospitals and ATI will also increase.

Finally, based on the theoretical analysis of Section 3.2, this paper adopts PNE to reflect whether there are transferring costs in the hospital [10,12]. Non-reimbursable expenses are services that cannot be reimbursed by medical insurance.

### 4.3. Statistical Analysis

To analyze the response of different-levels of hospitals to RGB-FPS, we divided the seven pilot hospitals into three levels, which is the classification recommended by WHO (Tertiary hospital, Secondary hospital, and Primary hospital) [31,32]. We analyzed the changes in the performance of tertiary, secondary, and primary hospitals before and after the reform of RGB-FPS.

For a more in-depth analysis of RGB-FPS’s impact, we focused on the changes in the number of inpatients, hospitalization cost, non-reimbursable expenses, and proportions of pilot hospitals at different levels through ITSA. The reason for selecting these types of indicators is that city A counts the monthly data of this type of indicator and uploads it to the information system.

ITSA is the strongest quasi-experimental method to estimate the post-policy changes in the level and trend of each outcome measure [33]. Segmented regression analysis is a powerful statistical method for estimating the intervention effects in interrupted time series studies. This method uses baseline trends and levels to project future monthly outcomes with the assumption that these values reflect what would have happened without the policy (i.e., the counterfactual). The basic model includes terms that estimate the baseline outline level (intercept), baseline trend (slope), change in the level of the outcome measured immediately after policy implementation, and change in post-policy trend. 

Segmented regression (with methods to account for autocorrelation) is the most commonly used modeling technique in ITSA. When only one group is under study (i.e., no comparison groups), the regression model is expressed as
$$Y_{t}{=\beta }_{0}{+\beta }_{1}\left(T\right){+\beta }_{2}\left(X_{t}\right){+\beta }_{3}\left({\mathrm{XT}}_{t}\right),\;$$
where $Y_{t}\;$ is the outcome variable during a period, which changes every month between July 2016 and July 2018, T is the time since the start of the study (July 2016 = 1, August 2016 = 2, …, December 2018 = 30), and $X_{t}$ is a dummy (indicator) variable that represents the intervention. Pre-intervention periods are denoted as 0; otherwise, the value is 1. In this study, the value of $X_{t}$ before July 2017 is 0, whereas that after this period is 1. ${\mathrm{XT}}_{t}\;$is an interaction term, which is 0 before July 2017, and then increases by 1 each month from July 2017 (1 = July 2017, 2 = August 2017, 3 = September 2017, …).${\;\beta }_{0}$ represents the intercept or starting level of the outcome variable before city A’s RGB-FPS, $\beta_{1}\;$is the slope or trajectory of the outcome variable until the introduction of city A’s RGB-FPS, $\beta_{2\;}$is the level change following the intervention, and $\beta_{3}$ indicates the slope change following the intervention. ${\mathrm{XT}}_{t}\;$is the interaction between time and intervention. The confidence interval of the *p*-value was 95%. 

For time-series data, seasonality is an issue that needs to be concerned. We handled the existence of seasonality through the moving average ratio method used in similar research [34]. Autocorrelation is another problem. Autocorrelation was assessed by examining the plot of residuals and the partial autocorrelation function, where data are normally distributed, conducting tests such as the BG test [35,36]. BG test suggested the existence of autocorrelations, which we corrected using the Regression with Newey-West standard errors [37].

### 4.4. Results

Table 1 and Figure 2 show the changes in outcome variables in different-level hospitals before and after RGB-FPS.

For tertiary hospitals, PI before intervention showed a downward trend (β_1_ = −0.033, *p* < 0.01), but after the implementation of RGB-FPS, PI showed a significant upward trend (β_3_ = −0.08, *p* < 0.001). Before the reform, ATI showed a stable fluctuation (*p* > 0.05), but ATI showed a significant upward trend after the implementation of RGB-FPS (β_3_ = 8.80 × 10^−4^, *p* < 0.001). PNE showed a downward trend before the reform (β_1_ = −0.074, *p* < 0.05), and there was no significant change in the trend after the reform (*p* > 0.05).

For secondary hospitals, PI times fluctuated steadily before the intervention, but after the implementation of RGB-FPS, PI decreased significantly (β_3_ = −0.074, *p* < 0.01). Before and after the reform, log(ATI) showed stable fluctuation (*p* > 0.05). PNE fluctuated steadily before the reform (*p* > 0.05) but increased significantly in the reform month (β_2_ = 1.07, *p* < 0.01) and showed stable fluctuation after the reform (*p* > 0.05).

For the primary hospital, PI before intervention showed a significant upward trend (β_1_ = 0.012, *p* < 0.05), but PI decreased significantly after the implementation of RGB-FPS (β_3_ = −0.014, *p* < 0.05). Log(ATI) stays fluctuated steadily (*p* > 0.05). PNE decreased significantly before the reform (β_1_ = −0.317, *p* < 0.01) but showed an obvious upward trend after the reform (β_3_ = 0.570, *p* < 0.001).

## 5. Discussion

Based on the empirical results, we tested our theoretical hypotheses: It is easier for large hospitals (tertiary hospitals) to increase service volume and for small hospitals (secondary hospitals and the primary hospital) to transfer costs.

Theoretically, due to the uncertainty of PV leading to the uncertainty of medical insurance income, all hospitals under RGB-FPS have the motivation to improve the service volume to occupy more medical insurance funds [4,12,27]. For doctors, the salary is also related to the amount of service volume. Therefore, the optimal strategy for doctors is to attract patients and hospitalization decomposition to increase service volume to maintain their salaries. Non-cooperative games among doctors also lead to negative hospital performance. From our empirical results, PI and ATI in large hospitals showed a significant upward trend in RGB-FPS (β3, *p* < 0.001). This result indicates that large hospitals increase service volume after implementing RGB-FPS, which is consistent with the empirical results of related studies [4,16,17]. However, transferring costs may result in higher out-of-pocket costs for patients and bring credibility problems to the hospital [12]. At the same time, large hospitals have resource advantages, so the optimal strategy of large hospitals is not to transfer costs. From the results, we can find that PNE in large hospitals did not change significantly (β3, *p* > 0.05). These results amply demonstrate that large hospitals are more likely to increase service volume than to transfer costs. 

Secondly, our empirical results show that after the reform of RGB-FPS, PI in small hospitals decreased, ATI did not change significantly, but PNE showed an upward trend. This suggests that small hospitals are negatively affected in terms of service volume and are more likely to transfer costs. Since human resources, equipment, and other resources of small hospitals are not as adequate as those of large hospitals, and the types of diseases in small hospitals are relatively simple, which makes it difficult for small hospitals to attract patients from large hospitals and hospitalization decomposition. Under RGB-FPS, the annual benefit of small hospitals is uncertain, and the hospital charges the patients according to FFS. Most people, including doctors, are averse to the risk of gains and prefer certain gains [38,39]. Therefore, small hospitals are willing to risk a loss of credibility to transfer costs to achieve a stable income. From the short-term influence of RGB-FPS, it is difficult to find a cooperative approach between small and large hospitals; therefore, some mandatory intervention strategies from policymakers, such as supervision of cooperation, might be required. It should be noted that RGB-FPS is targeted at all hospitals in the region and covers almost all inpatient categories. The effect of RGB-FPS on different diseases may be different. However, the empirical study of Shou-Hsia only included four kinds of diseases, which may lead to the lack of significant results [12]. 

Finally, this study verifies that RGB-FPS has a similar impact on the public-hospital-dominated market as it does on the private-hospital-dominated market. We think there could be two reasons: (1) the Chinese government has limited investment in public hospitals. Data show that government spending on public hospitals in China accounted for only 10.1% of total revenue in 2018, which means that public hospitals in China still have to pay for themselves [40]. Public hospitals in China will also take negative actions for profit when regulatory policies are inappropriate. (2) under the RGB-FPS mechanism, PV is uncertain, and not taking negative actions (such as over-provision of services) may result in loss of benefits. Most people are risk-averse, and public hospital administrators are no exception [38,39]. In order to maintain the share of medical insurance funds, the optimal strategy of different hospitals in the non-cooperative game is to take negative behavior.

This study has two major contributions: (1) We theoretically and empirically verified that different-levels hospitals respond differently to RGB-FPS. Large hospitals are more likely to increase service volume, and small hospitals are more likely to transfer costs. At the same time, this study also empirically verified the impact of RGB-FPS on the public-hospital-dominated market, which is similar to the private-hospital-dominated market. (2) The sample selection and methods of this study are complementary to the previous studies. First of all, the supervision system may affect the implementation performance of RGB-FPS [13]. The reform of RGB-FPS in many areas is accompanied by a change of the regulatory policy, so most studies cannot rule out the influence of other supporting policies and cannot simply analyze the effect of RGB-FPS. During our study period, using the seven pilot hospitals as samples can exclude the influence of other factors and facilitate us to discuss the simple influence of RGB-FPS. Secondly, the trend change in hospitals’ performance is not only affected by time trends but also by confounding factors [4]. Previous studies mostly used microscopic patient data. In this study, ITSA was used to control the impact of confounding factors on hospitals’ performance to some extent, and the impact of RGB-FPS on hospitals was verified from the perspective of macro-indicators.

### Limitations

Some limitations exist in this study. First, because of the availability of data, this study analyzed only 30-month changes in impact and was not able to analyze the long-term impact of RGB-FPS. Secondly, increasing service volume and transferring costs of hospitals may be affected by other factors besides RGB-FPS, such as the degree of market competition, government intervention, inter-hospital cooperation, and the connection between hospital managers, etc. This paper mainly focuses on the influence of different hospital levels on hospital behavior, and the influence of other factors cannot be excluded. Thirdly, RGB-FPS may influence the quality of hospital medical service. However, the literature has shown that RGB-FPS has different influences on the quality of medical service for different diseases, so the quality of service is not analyzed in this paper.

## 6. Conclusions

RGB-FPS is a strategy of cost control by increasing the degree of market competition. However, simply promoting market competition is difficult to achieve the desired effect [41]. The results of this study show that under the short-term influence of RGB-FPS, hospitals are driven by economic incentives and tend to take different negative behaviors, such as increasing service volume and transferring costs. Therefore, it is necessary to take some measures to control these negative effects. For example, Policymakers can take measures to avoid excessive competition between different hospitals and accelerate the progress of cooperation between hospitals. Policymakers should implement pay-for-performance payments to control possible negative effects. Different types of supervision and management measures should also be adopted for hospitals of different levels. For example, the excessive service provision of large hospitals can be supervised by PI, ATI, and other indicators, and the transferring cost of small hospitals can be supervised by PNE and other indicators.

## Figures and Tables

**Figure 1 ijerph-19-15507-f001:**
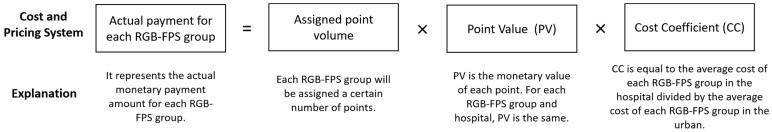
The core composition of RGB-FPS.

**Figure 2 ijerph-19-15507-f002:**
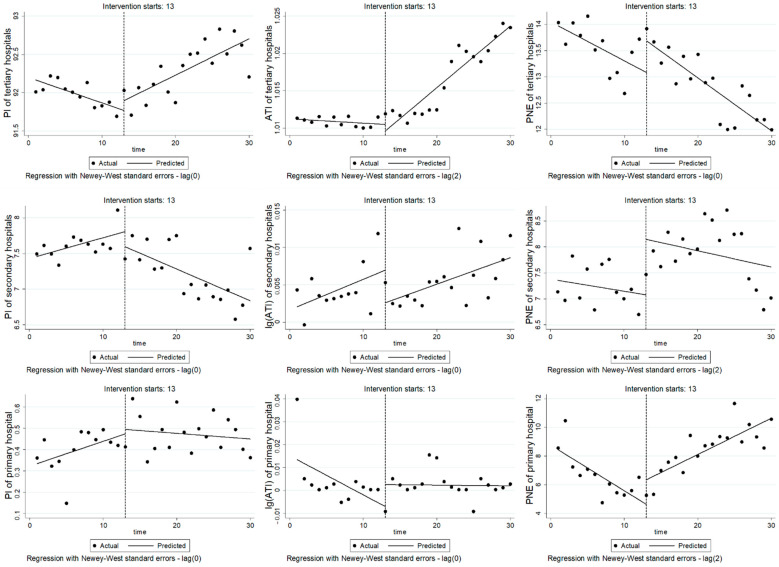
Figure of ITSA for outcome variables in different-levels hospitals. Note: The first, second and third lines are the index changes of tertiary hospitals, secondary hospitals and primary hospital, respectively.

**Table 1 ijerph-19-15507-t001:** Results of outcome variables in different-levels hospitals.

Hospital Level	Outcome Variables	β_1_, Level Change		β_2_, Level Change		β_3_, Trend/Slope Change	
		β_1_	(95% CI)	β_2_	(95% CI)	β_3_	(95% CI)
Tertiary hospitals	PI	−0.033 **	−0.053 to −0.012	0.128	−0.111 to 0.368	0.080 ***	0.048 to 0.112
	ATI	5.00 × 10^−5^	−1.30 × 10^−4^ to 1.97 ×10^−5^	−8.70 × 10^−4^	−0.003 to 0.001	8.80 × 10^−4^ ***	6.60 × 10^−4^ to 0.001
	PNE	−0.074 *	−0.142 to −0.006	0.602	−0.038 to 1.24	−0.027	−0.099 to 0.045
Secondary hospitals	PI	0.029	−0.005 to 0.063	−0.212	−0.593 to 0.170	−0.074 **	−0.122 to −0.026
	Log(ATI)	4.10 ×10^−4^	−2.56 × 10^−4^ to 0.001	−0.004	−0.010 to 0.001	−5.54 × 10^−4^	−7.59 × 10^−4^ to 6.49 × 10^−4^
	PNE	−0.023	−0.073 to 0.026	1.07 **	0.476 to 1.66	−0.008	−0.104 to 0.088
Primary hospital	PI	0.012 *	0.001 to 0.022	0.021	−0.097 to 0.139	−0.014 *	−0.028 to −3.40 × 10^−4^
	Log(ATI)	−0.002	−0.004 to 7.15 × 10^−4^	0.010	−0.004 to 0.023	0.002	−8.02 × 10^−4^ to 0.004
	PNE	−0.317 **	−0.526 to −0.109	1.71	−0.411 to 3.84	0.570 ***	0.358 to 0.781

Note: * *p* < 0.05, ** *p* < 0.01, *** *p* < 0.001. There are four tertiary hospitals, two secondary hospitals and one primary hospital.

## Data Availability

All data related to this study are included in the article or uploaded as Appendix A.

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
