# Peer review of "The Response of Different-Levels Public Hospitals to Regional Global Budget with a Floating Payment System: Evidence from China"

_ijerph, 2022, doi:10.3390/ijerph192315507_

Round 1
Reviewer 1 Report
Thank you for your manuscript, it is addressing a genuine knowledge gap.
- In the introduction: Regarding the Chinese context, where your study takes place, could you make a comment between urban/rural areas regarding RGB-FPS? There is a marked difference between urban/rural areas and their healthcare coverage in China and would be interesting to at least mention this angle.
- It is important in the discussion to consider whether (based on your findings) a co-operative approach is likely to emerge from smaller hospitals as part of the RGB-FPS introduction.
- Potential mitigation actions relating to the two main identified negative impacts are not mentioned or alluded to. Please link your findings (within a couple of brief sentences) to potential downstream activity to mitigate those negative impacts.
Author Response
For a clearer review, we have attached our responses to the reviewer's comments. Please see the attachment.

Reviewer 2 Report
The authors undertook the analysis of a very important, but at the same time extremely difficult research problem. Due to the limited resources in health care, rising costs of treatment and, at the same time, growing needs for treatment on the part of patients, the construction of an effective system of financing services becomes an urgent need not only in China.
However, determining the forces influencing the number of healthcare services and shifting their costs requires a precise determination of all factors. By the way, the study ignores the quality of services, which is also important in terms of the health status of individuals and society.
The authors proposed a preliminary and very simplified voice in the discussion on the structure of the health care financing system. They chose some factors that are important but omitted those that are considered more important, such as factors influencing treatment decisions by doctors. Using the FFS method for out-of-pocket payments is not a neutral condition and always winds up a spiral in the number of benefits.
In the part presenting the institutional background in Figure 1, a diagram is presented, which is at such a general level that probably everything can be included in the cost accounting.
Theoretical analysis is rather an intellectual exercise, or an introduction that is too distant from the conditions that hospitals encounter in their operation.
In the empirical part, the variables were the results of the decisions made earlier, and not the factors that led to them.
In my opinion, the results obtained in such a study are unreliable. Maybe the authors should precisely list all the limitations of the study, i.e. factors that have not been taken into account as determinants of changes in the number of benefits and cost transfer?
Author Response

(The authors gave the same response as above.)

Reviewer 3 Report
This study evaluated the regional global budget with floating payment system from the health providers’ perspectives. The findings might assist policymakers to make better decisions for the healthcare system. I would like to provide some major issues the authors should consider.
1. a more precious objective for this study needs to be clarified.
2. The parts in theoretical analysis needs to be refined and restructured. The readers will feel confused by the current structure and contents in that section.
3. the outcome variables in the Empirical analysis Section needs to be simplified and defined clearly.
Author Response

(The authors gave the same response as above.)

Round 2
Reviewer 2 Report
Descriptions in individual parts of the article have been changed and corrected. The study itself has not undergone any changes, so its relevance to practice is the same as the previous version. The answers contained in the appendix justify the authors' approach, but do not "refute" the allegations contained in the review.
Reviewer 3 Report
No comments